# Amplification-Free Testing of microRNA Biomarkers in Cancer

**DOI:** 10.3390/cancers17162715

**Published:** 2025-08-21

**Authors:** Bahareh Soleimanpour, Juan Jose Diaz Mochon, Salvatore Pernagallo

**Affiliations:** 1Destina Genomica S.L., Parque Tecnológico de la Salud de Granada (PTS), Avenida de la Innovación 1, Edificio BIC, 18016 Granada, Spain; bahareh@destinagenomics.com; 2Departamento de Química Farmacéutica y Orgánica, Facultad de Farmacia, Campus de Cartuja, 18071 Granada, Spain; juandiaz@go.ugr.es; 3Unidad de Excelencia de Química Aplicada a Biomedicina y Medioambiente, Facultad de Farmacia, Campus de Cartuja S/n, 18071 Granada, Spain; 4Centro Pfizer-Universidad de Granada-Junta de Andalucía de Genómica e Investigación Oncológica (GENYO), Avenida de la Ilustración 114, 18016 Granada, Spain; 5Instituto de Investigación Biosanitaria de Granada—ibs.GRANADA, Avenida de Madrid, 15, 18012 Granada, Spain

**Keywords:** microRNAs (miRNAs), PCR-free technologies, labeling-free detection, amplification-free platforms, cancer diagnostics, early detection, liquid biopsy, biomarker analysis, nucleic acid testing, sensor-based detection, multiplexed assays

## Abstract

Circulating microRNAs (miRNAs) show great promise as biomarkers for diagnosing and monitoring various diseases, including cancer. However, their use in clinical diagnostics is currently limited due to the lack of direct analytical methods. This review emphasizes the urgent need for amplification-free technologies to overcome these limitations. It explores recent innovations designed to streamline workflows, enhance accuracy, and expand the clinical applicability of miRNAs as diagnostic tools. Such advancements are essential for bridging the gap between research and clinical practice, as well as facilitating the development of more accessible and reliable miRNA testing tools.

## 1. Introduction

Cancer remains a significant global health challenge, causing almost 10 million deaths each year [1]. According to the American Cancer Society’s Cancer Statistics Update, over 2 million people were diagnosed with cancer in 2024 [2]. This results in around 5500 cases a day, or one diagnosis every 15 s. For the first time, the number of newly diagnosed cases in the United States has exceeded two million, with a growing proportion being identified at an earlier stage, when treatment success rates are at their highest. Consequently, cancer-related deaths have steadily declined, with an estimated 4.1 million lives saved since 1991 [3]. This progress reflects substantial investments in research and screening initiatives by organizations such as the National Institutes of Health, the Centers for Disease Control and Prevention, and the American Cancer Society [2,4].

Significant efforts continue to be focused on improving diagnostic and therapeutic methods in the ongoing fight against cancer. Early detection and precise diagnosis are crucial for improving patient outcomes and survival rates. Although tissue biopsy has long been considered the gold standard for cancer diagnosis, its invasiveness, incomplete tumor representation and risk of complications can limit its effectiveness [5].

Recent advancements in liquid biopsy, involving the detection of cancer biomarkers in bodily fluids such as blood, urine and saliva, have emerged as a transformative approach to non-invasive cancer detection [6,7]. Biomarkers such as circulating tumor cells (CTCs), circulating tumor DNA (ctDNA), microRNAs (miRNAs), and exosomes can provide real-time, comprehensive insights into tumor biology. Compared to traditional tissue biopsies, liquid biopsies offer clear advantages, including non-invasive and repeatable sampling, reduced risk, real-time monitoring and lower costs [7,8].

Furthermore, recent advances in molecular biotechnology have made it possible to identify cancer biomarkers with great precision through genomics, transcriptomics and proteomics [9]. These cutting-edge technologies have made a significant contribution to uncovering the underlying mechanisms of cancer and identifying actionable therapeutic targets, thereby accelerating the development of personalized and precision medicine approaches [10,11].

miRNAs are small non-coding RNA molecules, typically 18–24 nucleotides in length, which are present in all eukaryotic cells [12]. They play a crucial role in regulating key biological processes, including cell division, differentiation, apoptosis and numerous physiological and pathological pathways [13,14]. Alterations in miRNA expression, either up- or downregulation, have been widely associated with various human diseases, particularly cancer [15,16]. The importance of miRNAs in biology and medicine was further emphasized by the award of the 2024 Nobel Prize in Physiology or Medicine, which recognized their fundamental role in cellular regulation and disease [17].

Over the past two decades, extensive research has demonstrated the potential of circulating miRNAs as powerful biomarkers for cancer diagnosis, prognosis and therapy [18,19]. They have several clinical applications: they can signal the onset of cancer as early detection biomarkers; they can provide insight into disease progression and staging as prognostic indicators; and they can help evaluate therapeutic response and guide treatment decisions as predictive biomarkers. Therefore, monitoring miRNA expression profiles can offer clinicians a valuable tool for assessing how a patient responds to anticancer drugs and other interventions [20,21]. One of the major advantages of miRNAs is that they can be detected in various biofluids, including blood, urine and saliva [22]. This enables minimal or non-invasive sampling, making circulating miRNAs highly attractive for routine clinical testing and liquid biopsy applications [23]. They are notable for their remarkable stability in body fluids, remaining intact and detectable despite challenging biological conditions—an essential feature for reliable diagnostics [23,24,25].

Despite their enormous clinical potential, miRNAs present significant analytical challenges that limit their use in diagnostics. Although their presence in accessible biofluids, such as blood, urine and saliva, enables non-invasive sampling and supports the development of liquid biopsy tests, miRNAs are inherently difficult to detect and quantify precisely. Their short sequence length, low abundance and high sequence similarity among family members make specific and reliable measurement difficult [26]. As mentioned above, although circulating miRNAs demonstrate exceptional stability in body fluids—an important trait for diagnostic use—this alone does not resolve the technical complexities involved in their isolation, enrichment and analysis [16]. Addressing these issues is essential if we are to realize the full diagnostic and therapeutic potential of miRNAs [27].

Over the past two decades, several analytical platforms have been used to evaluate miRNA expression in biofluids, replacing conventional methods. These include the reverse transcription quantitative polymerase chain reaction (RT-qPCR) [28], droplet digital PCR (ddPCR) [29], microarrays [30] and next-generation sequencing (NGS)-based methods [31]. Using them for miRNA detection typically requires additional workflow steps, most notably reverse transcription to convert RNA into complementary DNA (cDNA), which can then be processed using standard DNA-based methodologies.

While these methods have demonstrated utility, they suffer from several limitations related to workflow complexity, cost and analytical accuracy [32]. Crucially, none of these technologies were originally designed for the analysis of miRNAs. Rather, they are adaptations of platforms developed for broader nucleic acid targets, such as messenger RNA (mRNA) and genomic DNA (gDNA) [33]. As a result, they are not ideally suited to the unique characteristics of miRNAs, including their small size (~22 nucleotides), absence of poly-A tails and high sequence similarity among family members. Moreover, circulating miRNAs are present in very low concentrations and are surrounded by complex backgrounds composed of other nucleic acids and abundant macromolecules. According to the literature, miRNAs constitute only about 0.01% of the total RNA mass in plasma [33,34]. Their expression levels can vary widely—from just a few copies to hundreds of thousands per microliter (μL) of blood plasma [35], corresponding to concentrations in the femtomolar (fM) to picomolar (pM) range.

These molecular features complicate probe design, increase the risk of off-target hybridization, and reduce analytical specificity and accuracy. This highlights the need for advanced analytical technologies with a dynamic range of at least four orders of magnitude. Furthermore, reliance on reverse transcription and amplification introduces technical variability and potential bias, which undermines reproducibility and limits the clinical translation of miRNA-based diagnostics.

These persistent limitations have prompted the scientific community to pursue the development of new, dedicated technologies optimized specifically for miRNA detection. Emerging approaches aim to enable direct interrogation of miRNAs in their native form, without requiring preprocessing steps such as RNA extraction, reverse transcription or amplification. One key goal is to achieve absolute quantification through robust, multiplexed, extraction-free assays comparable to the immunoassays used in protein biomarker analysis. These innovations are essential not only for improving analytical performance but also for accelerating the clinical translation of miRNA biomarkers for early disease detection, patient stratification and personalized therapeutic monitoring.

## 2. Conventional Analytical Methods

### 2.1. Quantitative and Digital PCR-Based Technologies

Among PCR-based methods, RT-qPCR is the most widely used and is considered the gold standard for detecting low levels of miRNA [36]. Originally developed for mRNA analysis, RT-qPCR offers high sensitivity for gene expression studies [37]. However, as stated above, applying it to miRNA analysis presents several challenges due to the unique molecular features of miRNAs: (1) Small size: The short length of miRNAs (typically 18–24 nucleotides) makes optimal primer and probe design difficult; (2) Structural overlap: Precursor miRNAs (pre-miRNAs) form stable hairpin structures, and mature miRNAs are derived from internal segments of these precursors, which complicates differentiation between the precursor and mature forms; (3) Lack of a poly-A tail: Unlike mRNAs, miRNAs lack poly-A tails, rendering them incompatible with the poly-T priming method employed in numerous traditional RT reactions; (4) Sequence similarity: Many miRNA family members (including isomiRs) differ by only one or a few nucleotides, which increases the risk of cross-reactivity and reduces specificity; (5) Low abundance: miRNAs are generally found in low concentrations in biofluids, which makes them highly susceptible to technical noise and contamination, particularly from gDNA. Therefore, efficient removal of gDNA before reverse transcription is critical for accurate results [32,38].

To enhance miRNA detection, RNA extraction kits that are specifically designed to enrich small RNAs (less than 200 nucleotides) are often used [39]. These help to exclude longer RNAs and improve specificity. However, such enrichment steps can also lead to the partial loss of target miRNAs, which could affect sensitivity if not carefully controlled.

RT-qPCR interrogates miRNAs by amplifying specific RNA sequences extracted from the biological matrix. The process begins with the extraction of total RNA, including miRNAs, from biological samples. The quality and purity of the extracted RNA significantly impact the accuracy and reliability of the subsequent analysis, so this step is critical. Following extraction, as shown in Figure 1, the RNA is reverse transcribed into cDNA [40]. This cDNA is then amplified in a solution containing DNA polymerase, nucleotides and primers that are complementary to the target DNA sequence. The amplification process involves three key steps: (1) Denaturation: The double-stranded DNA (dsDNA) is heated to separate it into single strands. (2) Annealing: As the solution cools, the primers bind to the target sequences on the separate DNA strands. (3) Extension: DNA polymerase adds nucleotides to the primers, creating complementary copies of the target DNA sequence. This cycle of denaturation, annealing and extension is repeated multiple times, exponentially increasing the quantity of the target DNA sequence [41]. Ideally, amplification does not occur in the absence of the target cDNA sequence, ensuring high specificity. Two commonly used detection chemistries in RT-qPCR are SYBR Green and TaqMan (Figure 1), each employing a distinct approach to detect amplified products [42]. SYBR Green is a fluorescent dye that binds to double-stranded DNA, emitting a signal proportional to the amount of dsDNA generated during amplification. In contrast, TaqMan uses a sequence-specific fluorescent probe, providing greater specificity and reducing the likelihood of non-specific amplification [42].

The combination of sensitivity and versatility has established RT-qPCR as a cornerstone technique for studying miRNA expression. Despite the challenges outlined above, it remains one of the most widely used and trusted methods for miRNA quantification and analysis.

Furthermore, digital PCR (dPCR) has emerged as a powerful PCR-based technique capable of absolute quantification, thereby addressing some of the limitations associated with traditional quantitative polymerase chain reaction (qPCR). In particular, ddPCR enables precise quantification by partitioning the sample into thousands of nanoliter-sized droplets, with each droplet functioning as an individual micro-reaction of defined volume [43]. After the process of PCR amplification, the fluorescence of each droplet is measured, and the droplets are classified as either positive (fluorescent) or negative (non-fluorescent) based on the presence or absence of the target nucleic acid. The number of positive droplets is then used to calculate the absolute concentration of the target using a Poisson distribution model [44]. These yields copy numbers per microliter of reaction along with associated confidence intervals [43].

A significant benefit of ddPCR is that it does not necessitate the use of reference genes or standard curves for quantification, thereby enhancing precision and eradicating variability introduced by amplification efficiency [45]. This is of particular benefit in the context of analysis of miRNAs, where the selection of a suitable reference gene can present significant challenges. In comparison to qPCR, ddPCR has been shown to possess several advantageous properties. These include an augmented level of sensitivity and accuracy in the detection of low-abundance targets, as well as an increased tolerance to polymerase chain reaction inhibitors. Consequently, ddPCR is regarded as a robust method for the analysis of complex or degraded samples [46].

**Figure 1 cancers-17-02715-f001:**
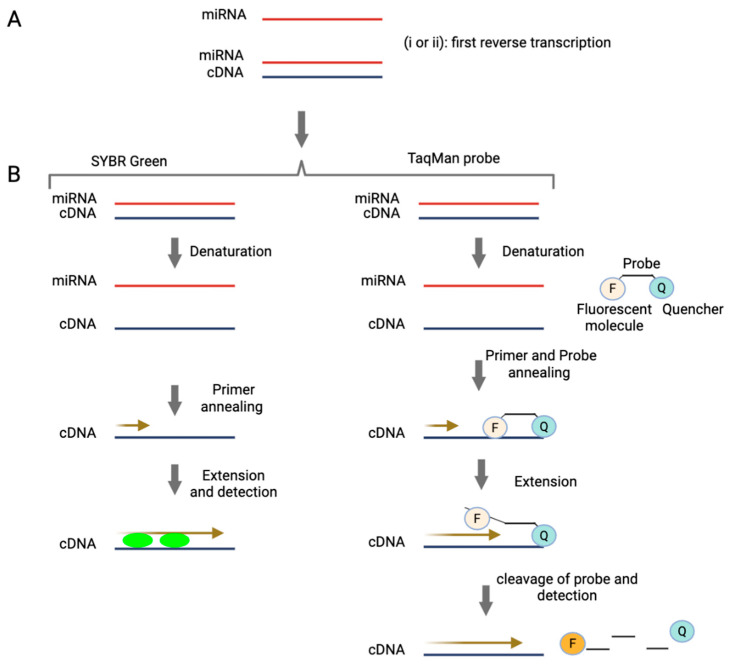
Methods of miRNA detection and quantification by RT-qPCR. (**A**) miRNA is first reverse transcribed to generate cDNA. This can be achieved using the following strategies: (i) Stem-loop priming: A stem-loop primer binds specifically to the target miRNA, initiating reverse transcription and providing high specificity [47]. (ii) The polyadenylation approach: In this method, a poly-A tail is added to the miRNA by an enzyme, enabling reverse transcription to proceed using an oligo(dT) primer [48,49]. (**B**) The resulting cDNA is then amplified by PCR and detected using one of two common methods: SYBR Green (or similar dyes): These intercalating dyes bind to double-stranded DNA during amplification, generating a fluorescent signal proportional to the amount of DNA. TaqMan probes: These sequence-specific probes consist of a fluorophore (F) and a quencher (Q). When the probe hybridizes to the target during PCR, the polymerase cleaves the probe, separating the fluorophore from the quencher and producing a fluorescence signal directly correlated to target amplification.

### 2.2. miRNA Microarrays

Microarray technology was originally implemented to study differences in transcription by analyzing mRNA, providing a robust method for examining gene expression profiles [50]. However, advancements in the field have enabled its use for detecting and quantifying short RNA molecules, such as miRNAs [51].

Microarrays are a high-throughput method that can be used to detect and measure changes in the levels of a wide range of miRNAs in a single experiment [52,53]. This technique is based on nucleic acid hybridization, whereby target molecules bind to their complementary probes that are anchored to a solid surface (e.g., glass slides) via covalent crosslinking [54]. Fluorescent dyes such as Alexa Fluor 546/647 or Cy3 are commonly used to label miRNAs. Those fluorescently labeled miRNAs then hybridize with the complementary probes on the microarray, resulting in specific binding. Detecting fluorescence emission at defined positions on the glass slide enables the evaluation of relative quantities of miRNAs in the sample by analyzing the intensity of the fluorescence signal (Figure 2). Microarray platforms enable the simultaneous comparison of expression levels in two different samples using distinct fluorophores. Numerous variants of microarray technology have been developed over the years for miRNA detection, incorporating innovations in immobilization chemistry, probe design, sample labeling and chip signal-detection methods [55]. Various commercial microarray platforms are now available for miRNA detection and quantification. However, studies have shown significant differences in their performance, including biases in miRNA quantification and their ability to determine expression profiles [56,57]. Despite their invaluable role in preliminary screening, microarrays are lacking in terms of the sensitivity and specificity necessary for absolute quantification [58].

### 2.3. Next-Generation Sequencing for miRNA Profiling

NGS has profoundly advanced the field of miRNA analysis, providing a powerful tool for high-throughput, comprehensive profiling of small RNA populations. This technology not only enables accurate quantification of known miRNAs but also supports the discovery of novel miRNA species.

The typical workflow for small RNA sequencing (sRNA-Seq) begins with the extraction of total RNA from biological samples (Figure 3). This RNA pool contains a variety of molecules, including miRNAs and other small non-coding RNAs. Following RNA extraction, a size selection step is performed to enrich the sample for small RNAs, generally in the range of 18 to 30 nucleotides. This selection is essential to exclude longer RNA molecules and to focus the sequencing effort on the small RNA fraction.

Once the small RNAs are enriched, synthetic adapters are ligated to their 3′ and 5′ ends. This adapter ligation is a critical step because it prepares the RNA molecules for reverse transcription and subsequent amplification. After adapter ligation, the small RNAs are reverse transcribed to generate cDNA, which is then amplified by PCR to produce a sequencing-ready library (Figure 3). The amplified libraries are sequenced, and the resulting reads are mapped to reference genomes or curated miRNA databases, such as miRBase, to identify and quantify known miRNAs and to explore the presence of potentially novel [59].

Library preparation can be performed using commercially available kits, which vary based on the sequencing platform and the specific needs of the experiment. Some protocols use adapters with fixed sequences, as commonly found in kits like Illumina TruSeq, NEBNext, and CleanTag. Other protocols, such as those provided by the NEXTflex system, use adapters that contain four randomized nucleotides at their ligation ends [60]. These randomized or “4N” adapters are specifically designed to minimize ligation bias by reducing sequence dependency during adapter binding, thereby improving the uniformity and efficiency of miRNA capture.

One of the key technical challenges in sRNA-Seq is the introduction of ligation bias. The sequence and secondary structure of miRNAs can significantly affect how efficiently adapters are ligated to the RNA molecules. This can lead to preferential ligation of some miRNAs and under-representation of others, distorting the actual abundance of miRNAs in the sample. Ligation bias is particularly problematic in small RNA sequencing and is generally more pronounced than in standard mRNA sequencing protocols [61].

Once sequencing is complete, the data analysis presents its challenges. In sRNA-Seq experiments, particularly when analyzing biofluids, miRNA expression levels are often highly skewed. A small number of highly abundant miRNAs typically account for the majority of sequencing reads, while most other miRNAs are present at low levels. This uneven distribution complicates data processing, especially in the normalization phase.

Different normalization methods have been proposed to address this challenge [62]. Some approaches adjust the data based on the total number of reads per sample, while others focus on aligning the distribution of expression levels across samples to improve comparability. Methods such as Reads Per Kilobase of transcript per Million mapped reads (RPKM) are widely used in mRNA sequencing, but they are less appropriate for miRNA sequencing because miRNAs are uniformly short and the RPKM adjustment is not meaningful for such molecules [63].

There is currently no universal consensus on the best normalization strategy for sRNA-Seq data. The choice of method can significantly influence the results of downstream analyses, including differential expression studies. Therefore, researchers must carefully select and justify their normalization approach based on the specific type of sample, experimental design and biological question being addressed.

Over the years, significant improvements in library preparation, adapter design and data normalization have helped to enhance the accuracy and reliability of miRNA profiling by NGS [64]. Nevertheless, careful attention to protocol selection, optimization steps and data interpretation remains essential to obtain high-quality and biologically meaningful results.

## 3. Aim and Methodology

Current approaches for analyzing circulating miRNAs typically involve RNA extraction and small RNA enrichment, steps that often result in partial miRNA loss and hinder the detection of low-abundance targets. Moreover, widely used methods such as RT-qPCR, ddPCR, microarrays and NGS rely on reverse transcription and amplification, which can introduce variability and compromise the accuracy of quantification [65]. 

These limitations highlight the need for direct detection strategies that bypass RNA extraction, reverse transcription and amplification to enable more robust and reproducible analysis of circulating miRNAs.

To address this need, we conducted a comprehensive literature review to identify technologies capable of detecting cancer-associated circulating miRNAs without requiring nucleic acid extraction or amplification. The search targeted original and review articles published up to 2024 in the PubMed and Scopus databases, using the following query:

((microRNA [Title/Abstract] OR miRNA [Title/Abstract]) AND amplification-free [Title/Abstract] OR PCR-free [Title/Abstract] AND detection [Title/Abstract]).

The search retrieved 459 manuscripts. After removing 344 manuscripts—comprising 75 duplicates and 269 published before 2020—115 manuscripts remained for screening. Of these, 87 were excluded based on the following criteria: 33 did not involve clinical samples, 37 were not focused on circulating miRNAs, and 17 were review articles. This left 28 manuscripts for full-text evaluation.

Among these, 19 were further excluded: 8 did not implement extraction-free methodologies, and 11 lacked essential technical or clinical parameters. 

Ultimately, 9 manuscripts met the inclusion criteria. 4 of these described distinct approaches not discussed in detail here, including the following:
A fluorescent spherical nucleic acid (FSNA)-assisted microfluidic chip developed for miRNA detection [66].A dual-aptamer modified gold nanoparticle (AuNP) system enabling universal miRNA detection via dye-fading readout [67].A cascade CRISPR/Cas-based method for amplification-free miRNA sensing [68].An ultra-fast, clickable, fluorescence-based detection strategy [69].

The remaining 5 emerging amplification-free methods for circulating miRNA detection in cancer, which form the core focus of this review, are described in detail in Section 4 and summarized in Table 1. These platforms represent diverse, state-of-the-art approaches with high sensitivity, proven clinical relevance and the capability to directly analyze biological fluids. Notably, one of them is the Dynamic Chemical Labeling (DCL) method, developed by the authors’ research group, which employs synthetic abasic Peptide Nucleic Acid (PNA) probes and chemically reactive nucleobases for direct, extraction-free, amplification-free detection of circulating miRNAs.

## 4. Emerging Amplification-Free Methods for Circulating miRNA Detection in Cancer

### 4.1. Solid-State Nanoplasmonic Sensor

Masterson et al. [70] used a solid-state nanoplasmonic sensor to detect circulating miRNAs, specifically miR-10b-5p and miR-let7a-5p, in plasma samples. This approach is both amplification-free and label-free. These miRNAs show great promise in the early detection of pancreatic ductal adenocarcinoma (PDAC) [75,76].

This sensor utilizes the unique localized surface plasmon resonance (LSPR) properties of gold triangular nanoprisms (Au TNPs) attached to the glass bottom of a 384-well plate [77,78]. Fabrication involves two key steps: (i) Nanoparticle immobilization: Chemically synthesized Au TNPs (42–55 nm edge length) are covalently attached to a mercaptomethoxysilane-modified glass substrate to ensure stable anchoring. (ii) Covalent functionalization: The Au TNPs are functionalized with thiolated single-stranded DNA (ssDNA) capture probes specific to the target miRNAs. Polyethylene glycol thiolate (PEG4) spacers are co-immobilized to minimize nonspecific binding and fouling (Figure 4).

The Au TNPs act as plasmonic nanoantennas, concentrating the local electromagnetic field at their surface and creating near-field “hot spots” upon LSPR excitation. The sensitivity of this system is highly dependent on the size, shape and local dielectric environment of the Au TNPs [79]. These near-field effects significantly enhance the sensor’s ability to detect minute refractive index changes upon miRNA hybridization.

The detection mechanism relies on the LSPR shift (Δλ_(LSPR)), which occurs when target miRNAs hybridize with the ssDNA probes, as also reported in [80]. Formation of a ssDNA/RNA duplex increases the local refractive index, resulting in a measurable red shift in the LSPR spectrum. This shift is quantified via UV–vis extinction spectroscopy [81]. This spectral shift is directly proportional to the concentration of the miRNAs, enabling accurate, label-free quantification.

In an earlier development involving the 96-well format [82], this solid-state nanoplasmonic sensor could detect miRNAs from as little as 10 µL of plasma with a LoD of 10^−18^ M (attomolar). To improve scalability, this work optimized the platform for a 384-well format, significantly enhancing sensitivity and throughput. In this configuration, the sensor achieved a LoD of 637.7 aM and a quantification limit of 45 fM.

The sensor exhibited exceptional diagnostic performance, effectively distinguishing PDAC from chronic pancreatitis (CP) and healthy controls, surpassing the accuracy of CA19-9, the current clinical biomarker standard [83,84]. Combined biomarker strategy using miR-10b-5p and miR-let7a-5p achieved approximately 91% sensitivity and 87% specificity in distinguishing early-stage (I/II) from late-stage (III/IV) PDAC [85], demonstrating strong potential for the early detection of cancer.

Despite these promising results, there are still challenges to overcome, including the need for overnight assay incubation, ensuring sensor stability, and conducting large-scale clinical validation. Future improvements should focus on achieving greater multiplexing capabilities to enable the simultaneous detection of multiple miRNAs, optimizing surface chemistry to reduce non-specific binding and integrating the technology into point-of-care diagnostic devices. Validation in larger and more diverse patient cohorts is essential to establish its clinical applicability. Nevertheless, this solid-state nanoplasmonic sensor is a significant advancement in non-invasive cancer diagnostics, offering a highly sensitive, specific and scalable solution for early detection, disease monitoring and personalized medicine.

### 4.2. Electro-Optical Nanopore Sensing

In this study, Cai et al. [71] used an electro-optical nanopore sensing platform for the direct detection of miRNAs from serum samples in a label-free, amplification-free, multiplexed approach. This single-molecule technique combines size-encoded molecular probes, nanopore electrical sensing and fluorescence microscopy. This enables highly sensitive and specific quantification of miRNAs with minimal sample input.

The detection mechanism is based on dual-mode signal readout, combining electrical nanopore sensing to discriminate between molecular probes based on size, with fluorescence-based optical detection to recognize target miRNAs based on sequence. This approach enables multiple miRNAs to be detected simultaneously in a single sample, making it a powerful tool for high-throughput liquid biopsy applications.

As shown in Figure 5, the system uses custom-designed molecular probes, each of which consists of a DNA carrier encoded molecular probe and a molecular beacon (MB). The MB is a stem-loop DNA structure labeled with a fluorophore and a quencher. It is designed to remain closed in the absence of the target miRNA, which prevents fluorescence emission. Upon hybridization with its complementary miRNA, the MB unfolds, separating the fluorophore from the quencher and restoring fluorescence for optical detection.

At the same time, the DNA carrier acts as a molecular barcode, enabling size differentiation using nanopores. Each probe–miRNA complex is driven through a solid-state nanopore by electrophoresis, generating distinct current blockade signals that depend on the length of the DNA carrier. Since each miRNA-specific probe is attached to a carrier of a unique length, nanopore analysis can distinguish between multiple miRNAs based on their electrical signatures. Integrating electrical and optical readouts enhances specificity, enabling the effective discrimination of single-base mismatches and improving the accuracy of miRNA profiling. Figure 5 shows the electro-optical nanopore sensing workflow.

This technology was validated using prostate cancer (PCa)-associated miRNAs, including miR-141-3p and miR-375-3p, which are significantly upregulated in PCa patients [86,87]. The assay achieved an ultra-low LoD of 5–8 femtomolar (fM), surpassing the sensitivity of RT-qPCR and fluorescence-based bulk assays. Moreover, a three-miRNA panel (miR-141-3p, miR-375-3p, and let-7b) enabled classification of localized versus metastatic PCa, demonstrating its potential for disease staging and treatment monitoring [88,89]. The system exhibited an accuracy of 98.8% in differentiating PCa patients from controls, underscoring its clinical relevance.

A key advantage of this nanopore sensing platform is its minimal sample requirement—only 0.1 µL of unprocessed serum—making it a minimally invasive and efficient tool for clinical diagnostics (Figure 5). Unlike PCR-based methods, which are prone to variability due to amplification bias, this amplification-free and extraction-free approach ensures more reliable quantification of miRNAs in biofluids. Additionally, the ability to perform multiplexed detection using size-encoded DNA carriers significantly enhances the system’s throughput and scalability, making it suitable for high-throughput screening applications [71].

Despite its advantages, several challenges remain. Further clinical validation across larger and more diverse patient cohorts is essential to confirm the platform’s diagnostic utility. Additionally, expanding multiplexing capacity to detect a broader range of cancer-associated miRNAs and optimizing the platform for point-of-care applications would enhance its clinical feasibility. Future developments should focus on automating sample processing, increasing detection speed, and integrating the system into hospital workflows to facilitate real-world implementation.

### 4.3. Singlet Oxygen-Based Photoelectrochemical

The singlet oxygen (^1^O_2_)-based photoelectrochemical (PEC) sensor used by Shanmugam et al. [72] represents a promising advancement in the detection of circulating miRNAs, providing a label-free, amplification-free, and highly sensitive strategy for early cancer diagnosis [90]. Unlike conventional methods that require complex reagents, enzymatic reactions, or nucleic acid amplification, this PEC system uniquely uses air as the oxygen source, significantly reducing costs and simplifying sensor fabrication and operation [91].

As shown in Figure 6A, the detection relies on a sandwich hybridization assay in which the target miRNA is specifically captured between two complementary DNA probes: a biotinylated capture probe immobilized on streptavidin-coated magnetic beads and a detection probe labeled with Chlorin e6 (ChlE6), a photosensitizer [92]. After hybridization, the magnetic beads are drawn to the sensor surface using an external magnetic field, ensuring the proximity of the probe-miRNA complex to the electrode for efficient signal generation [92,93].

When the sensor is exposed to light, the photosensitizer produces singlet oxygen (^1^O_2_), which rapidly reacts with the redox mediator hydroquinone (HQ), oxidizing it to benzoquinone (BQ). The BQ is then electrochemically reduced at the electrode back to HQ, establishing a redox cycling loop that greatly amplifies the PEC signal (Figure 6B).

Optimization of the system involved fine-tuning the electrode potential to −0.2 V vs. an Ag quasi-reference electrode, adjusting the number of magnetic beads to 100 µg per assay and the HQ concentration to 1 mM. This resulted in a LoD of 0.62 pM for miR-141-3p in buffer. The method demonstrated high specificity, low background signal and robust performance in untreated plasma samples, highlighting its potential for clinical use. Its simplicity, cost-effectiveness and rapid response time make it ideal for point-of-care diagnostics. The clinical applicability of this PEC platform was further validated by analyzing circulating miRNAs in plasma samples from prostate cancer patients [94]. Following assaying of buffer samples, detection limits of 3.5 pM for miR-145-5p and 8.3 pM for miR-141-3p were established in pooled plasma spiked with target miRNAs at varying concentrations. Finally, when the platform was used with plasma samples from eight prostate cancer patients, significantly higher photocurrent responses (8–18 nA) were detected compared to healthy controls (4.5–5.5 nA). This is consistent with the elevated levels of miR-145-5p and miR-141-3p observed in prostate cancer [95,96]. Notably, miR-145-5p showed more consistent elevation across samples than miR-141-3p. This method enables sensitive and specific miRNA detection directly in untreated plasma, eliminating the need for pre-amplification and simplifying data interpretation. This makes it particularly attractive for point-of-care applications in clinical biomarker analysis [97].

### 4.4. Tandem Bead-Based Hybridization Assay

The tandem bead-based hybridization assay used by Slott et al. [73] offers a novel, amplification-free strategy for both detecting miRNAs and performing single-nucleotide polymorphism (SNP) analysis directly in miRNA sequences. This dual capability enables sensitive and specific quantification of circulating miRNAs while simultaneously distinguishing single-base mutations, which are increasingly recognized as important biomarkers in diseases such as colitis and colorectal cancer [98,99].

The detection system is based on a two-step hybridization assay that uses locked nucleic acid (LNA)-enriched probes to maximize binding affinity and SNP discrimination [100]. The assay workflow is summarized in Figure 7. Briefly, in the first step, biotinylated, mutation-specific capture probes (C1-C14) are immobilized on streptavidin-coated magnetic beads, selectively binding either wild-type or mutant miRNA sequences. In the second step, a linker probe—designed using the Peyrard–Bishop mesoscopic model—binds to the 5′ end of the miRNA and incorporates a calf thymus DNA (CTD) booster sequence to enhance the fluorescence signal.

The detection is achieved using QuantiFluor dye, which was selected after systematic evaluation of five fluorophores (EvaGreen, AccuClear, QuantiFluor, Acridine Orange and Thiazole Orange) for photostability and emission performance [101]. QuantiFluor provided the most stable signal with minimal photobleaching (2.5% over seven cycles) and enabled the assay to achieve a LoD of 2.2 pM.

The assay was validated using plasma samples from 20 colorectal cancer patients, 24 patients with colitis, and 20 healthy controls. It successfully identified distinct SNP profiles in miR-128-2-3p that correlated with disease status [98,99]. Notably, mutations at the 3′-2 position (U > G and U > C) were found to be significantly elevated in diseased samples, suggesting their potential as biomarkers for colorectal cancer and colitis.

This technology offers several key advantages: (a) It enables simultaneous miRNA detection and SNP analysis in a single assay. (b) It is completely enzyme-free and amplification-free, reducing assay complexity, cost, and technical variability. (c) It provides high specificity and discrimination for SNPs, with melting temperature differences (ΔTm) ranging from 6.6 °C to 25.5 °C between perfectly matched and mismatched sequences. (d) The bead-based format supports multiplexing and high-throughput screening, making it suitable for large-scale clinical applications. (e) Compared to traditional enzymatic assays, it significantly simplifies the workflow while delivering robust sensitivity and reproducibility.

Overall, this tandem hybridization assay is a powerful and cost-effective platform for directly detecting miRNAs and their SNP variants from plasma. It has strong potential for clinical translation in the areas of early cancer diagnosis and personalized medicine. However, further refinements are required to improve its multiplexing capabilities, automate sample processing and validate its performance across different patient groups. Nevertheless, the assay remains a promising amplification-free approach for non-invasive cancer diagnostics, providing a rapid, sensitive and scalable tool for miRNA mutation profiling in clinical settings.

### 4.5. Dynamic Chemical Labeling

Dynamic Chemical Labeling (DCL) is a chemical-based nucleic acid testing (NAT) strategy developed to directly detect and quantify circulating miRNAs without the need for extraction, reverse transcription or enzymatic amplification [102]. Unlike conventional techniques, DCL relies on a sequence-selective chemical labeling process that provides exceptional specificity, single-nucleotide discrimination and direct detection in complex biological fluids [103].

As shown in Figure 8, the DCL detection process involves two highly controlled molecular steps. First, an abasic PNA probe selectively hybridizes to the complementary miRNA sequence, forming a chemical pocket at the abasic site. Once hybridization is complete, a SMART base with a specific labeling tag, such as biotin, is covalently incorporated into the abasic site. This creates a chemical lock that tags the abasic PNA probe-miRNA duplex for detection. The labeled duplex is then recognized by a reporter molecule—typically a streptavidin-conjugated detection system—enabling signal generation across a variety of platforms [103,104,105,106,107,108]. A key feature of DCL is its dual molecular requirement for signal generation: (a) Perfect hybridization between the miRNA and the abasic PNA probe as well as (b) Selective molecular recognition and covalent incorporation of the SMART base following Watson–Crick base-pairing rules. If either event fails, the SMART base is not incorporated, resulting in 100% specificity [107,109].

This dual specificity drastically reduces false positives and allows single-nucleotide specificity, a rare feature in amplification-free miRNA detection systems [104,110,111]. The labeled abasic PNA probe–miRNA duplexes can be detected using various optical and bead-based platforms, making DCL highly versatile [112,113,114].

Importantly, the sensitivity of DCL is not intrinsic to the chemistry itself but is rather determined by the detection platform with which it is integrated. DCL is a versatile technology that can be successfully integrated with various platforms, including fluorescence-based and chemiluminescence detection systems, as well as silicon photomultiplier (SiPM)-based readers and time-gated luminescence imaging systems [103,105,108,115,116]. This flexibility allows the sensitivity, throughput and multiplexing capability of the system to be adjusted according to the specific platform, application and target RNA or DNA [110,114,117,118,119]. It also enables the simultaneous detection of miRNAs and proteins associated with liquid biopsy [115,120].

The group successfully applied DCL technology to analyze circulating hsa-miR-21-5p (miR-21) in patients with non-small cell lung cancer (NSCLC) [121]. To achieve this, DCL was combined with a novel SiPM-based optical reader, resulting in the development of the innovative ODG platform [122]. This method utilized biotinylated SMART bases and abasic PNA probes specifically designed to capture miR-21 sequences, enabling highly selective, sequence-specific miRNA detection. The captured and labeled miRNAs were detected through a chemiluminescent reaction, in which streptavidin-conjugated horseradish peroxidase (HRP) catalyzed the luminol oxidation, generating a light signal. This signal was subsequently measured and analyzed using the SiPM-based reader. The platform achieved a LoD of 4.7 pmol/L and successfully identified miR-21 in plasma samples from NSCLC patients. Although this approach provided accurate, amplification-free quantification of miRNAs, it faced challenges related to signal stability and background noise, which are inherent limitations of chemiluminescence-based detection systems. miR-21 was also interrogated using DCL technology merged with flow cytometry in a PCR-free manner as reported in [123].

In 2024, a multiplexed miRNA detection platform was developed that integrated DCL with time-gated photoluminescence imaging (TG-PLIM) to enhance signal and significantly reduce background interference. Similar to the previously developed ODG platform, this system employed abasic PNA probes and SMART bases. However, instead of using chemiluminescence detection, the platform was adapted to a fluorescence-based detection system that utilized fluorophores with long luminescence lifetimes, specifically lanthanide-based luminescent probes such as Eu(III) cryptates conjugated to the SMART base [74].

In this novel application of DCL, the system was designed to simultaneously detect three clinically relevant miRNAs involved in cancer diagnostics: (a) miR-122-5p (miR-122), increasingly recognized as a potential biomarker for cancer diagnosis and prognosis due to its dysregulation in various cancers [124], (b) miR-371a-3p (miR-371), a well-validated biomarker for germ cell tumors, commonly used in both clinical and research settings [125], and (c) miR-451a-5p (miR-451), an erythroid cell-specific miRNA with emerging relevance as a biomarker in cancer diagnostics and therapeutic response monitoring [126].

In this study, magnetic beads functionalized with specific abasic PNA capture probes (capturing beads) were combined with SMART bases labeled with DTBTA-Eu cryptates and conventional fluorophores such as FAM and Cy5, enabling the simultaneous detection of three miRNA targets within the same assay (Figure 9A). A key advantage of using DTBTA-Eu cryptates is their long luminescence lifetime of 1.09 ms, which allows error-free multiplexing by separating signal detection into distinct time windows, effectively preventing photon crosstalk [127]. Following bead incubation, the DCL reaction, and time-resolved imaging, spectral and lifetime filtering were applied to differentiate each miRNA based on its distinct emission properties (Figure 9B,C). TG-PL intensity images of individual beads demonstrated a concentration-dependent detection of miRNAs, with a linear correlation between TG-PL emission intensity and miRNAs concentrations.

The system achieved a LoD of 1.4 nM for miR-122, validating its sensitivity for detecting clinically relevant miRNA levels in human serum. The combination of spectral and temporal separation enabled the simultaneous detection of multiple miRNAs in a single assay, each assigned a unique fluorescent label. The labeled beads were further analyzed using machine learning algorithms, which classified and quantified the detected miRNAs with high accuracy and minimal misclassification.

This study successfully demonstrated the proof of concept for a multiplexed miRNA detection platform using DCL and TG-PLIM. The platform exhibited high specificity and the ability to simultaneously detect three cancer-related miRNAs. While the achieved LoD of 1.4 nM confirms the platform’s clinical potential, further improvements are essential to reach the femtomolar to picomolar range typical of circulating miRNAs. The group is currently working to enhance sensitivity to unlock the full diagnostic potential of the platform.

## 5. Conclusions and Future Direction

Conventional technologies such as qRT-PCR, microarrays and NGS are widely used for miRNA analysis. However, their application to the detection of circulating miRNAs in clinical settings presents several challenges. These techniques often require complex and labour-intensive workflows involving RNA extraction, enzymatic amplification and labeling steps, all of which introduce potential sources of bias, increase contamination risks, extend turnaround times and drive up costs. Their limited scalability and technical demands also hinder their integration into routine diagnostics or point-of-care applications.

In response to these limitations, amplification-free detection platforms have emerged as promising alternatives. The technologies reviewed in this article—including solid-state nanoplasmonic sensors, electro-optical nanopore sensing, singlet oxygen-based photoelectrochemical detection, tandem bead-based hybridization assays and DCL—enable the direct detection and quantification of circulating miRNAs in biological fluids without the need for RNA extraction, reverse transcription or enzymatic amplification. This represents a significant advancement in liquid biopsy diagnostics, offering enhanced specificity, reduced variability and simplified sample processing. Many of these methods are compatible with multiplexed and bead-based formats, making them adaptable to diverse clinical workflows and potentially suitable for decentralized testing environments.

Crucially, these platforms offer multiple advantages for clinical translation. By eliminating the need for multiple processing steps, they streamline workflows, shorten assay times and reduce operational complexity. Several of the reviewed technologies, including DCL and nanopore-based methods, can be adapted for partial or full automation using closed-cartridge or bead-based systems, enabling standardization and minimizing user input. Moreover, reduced reliance on costly enzymes and reagents lowers the per-test cost, an important consideration for widespread screening and healthcare adoption.

Each of the technologies discussed offers distinct features, strengths and limitations. Their potential for multiplexing, label-free detection and compatibility with various readout platforms support a range of use cases across different stages of cancer diagnostics. Some platforms focus on single-analyte precision, while others are designed for high-throughput or point-of-care testing. Collectively, they reflect a growing shift toward streamlined, clinically adaptable solutions for miRNA-based diagnostics.

The clinical value of miRNA biomarkers was further underscored by the 2024 Nobel Prize in Medicine, which recognized the transformative potential of circulating nucleic acids in non-invasive diagnostics. miRNAs are now well established as key regulators of cancer-related pathways, with distinct expression profiles across tumor types, disease stages and therapeutic responses. Technologies capable of reliably detecting these biomarkers in their native form—without reverse transcription or amplification—are essential for real-time, non-invasive diagnosis, prognosis and patient stratification.

Beyond miRNAs, amplification-free detection technologies also hold promise for broader applications, including the analysis of other RNA species such as circular RNAs (circRNAs) and long non-coding RNAs (lncRNAs), without the need for pre-processing steps [128,129]. Ongoing research within our group is actively exploring these extended capabilities.

Looking forward, the continued refinement of user-friendly, scalable and highly specific amplification-free platforms will be critical to advancing the clinical adoption of liquid biopsy diagnostics. Technologies that combine rapid turnaround, minimal sample input and accurate detection of low-abundance targets in complex matrices are poised to shape the future of molecular diagnostics. Ultimately, these platforms could redefine standard practice in cancer diagnostics, enabling earlier disease detection, improved monitoring and more personalized therapeutic strategies.

## Figures and Tables

**Figure 2 cancers-17-02715-f002:**
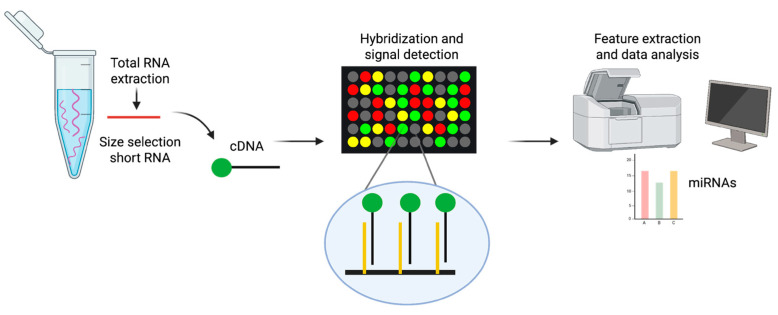
Diagram of the miRNA microarray workflow. From left to right: Size selection is used to enrich the total RNA for short RNA, including miRNA. Labeling is typically performed by incorporating a fluorophore during the reverse transcription and cDNA generation steps. For simplicity, the figure shows only one probe for a type of miRNA; however, multiple identical probes are present in clusters on the microarray surface and are designed to bind to identical target miRNAs. A complete microarray contains thousands of clusters, with up to 20 repetitions of the same cluster distributed across different locations on the slide. Wash steps are omitted for clarity.

**Figure 3 cancers-17-02715-f003:**
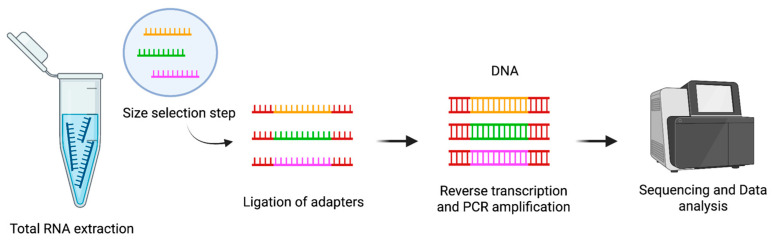
Diagram of small RNA Sequencing workflow. From left to right: The process begins with the isolation of small RNAs, followed by the ligation of synthetic adapters to their 3′ and 5′ ends—a crucial step in enabling reverse transcription. The adapter-ligated RNAs are then converted into complementary DNA (cDNA) and amplified by polymerase chain reaction (PCR) to generate a sequencing-ready library. This method enables the detection of thousands of small RNA molecules, including both known and novel species. The workflow details may vary depending on the library preparation protocol and sequencing platform used. For simplicity, only three small RNA types are indicated (colored miRNAs).

**Figure 4 cancers-17-02715-f004:**
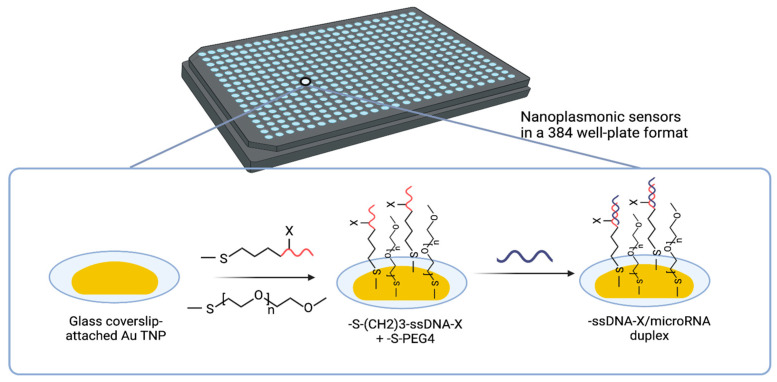
Solid-state nanoplasmonic sensor for detection and quantification of miRNAs. The platform is based on Au TNPs immobilized on the glass surface of a 384-well plate. These Au TNPs are functionalized with thiolated ssDNA capture probes, which specifically hybridize with target miRNAs in the plasma sample. The binding event forms an ssDNA/miRNA duplex at the sensor surface, inducing a LSPR spectral shift due to changes in the local dielectric environment. The PEG4 spacers are included to minimize nonspecific binding and fouling effects. Figure adapted with permission from [70].

**Figure 5 cancers-17-02715-f005:**
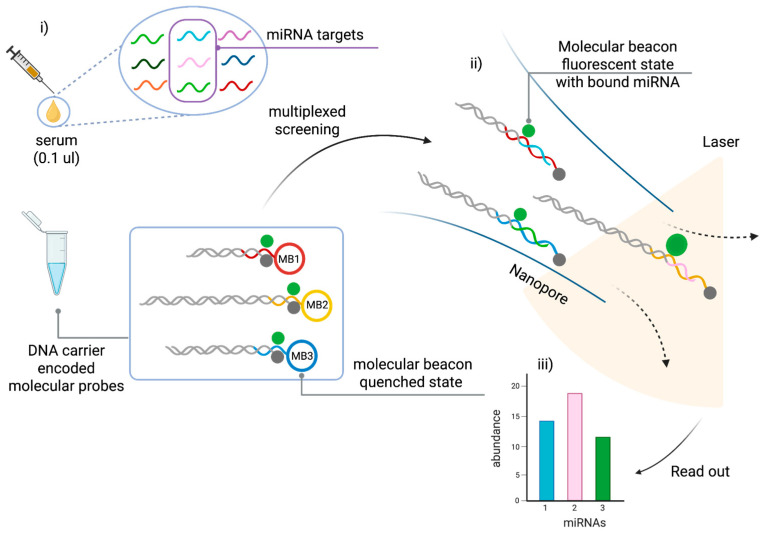
Single-molecule multiplexed sensing using an electro-optical nanopore platform: (i) Patient serum is incubated with a length-encoded molecular probe consisting of a DNA carrier and a molecular beacon (MB). (ii) Electro-optical sensing is performed. (iii) MiRNA expression levels are determined. Figure adapted with permission from [71]. Figure 1B of [71] shows a schematic representation of the preparation of size-coded DNA probes and their binding to respective miRNA targets.

**Figure 6 cancers-17-02715-f006:**
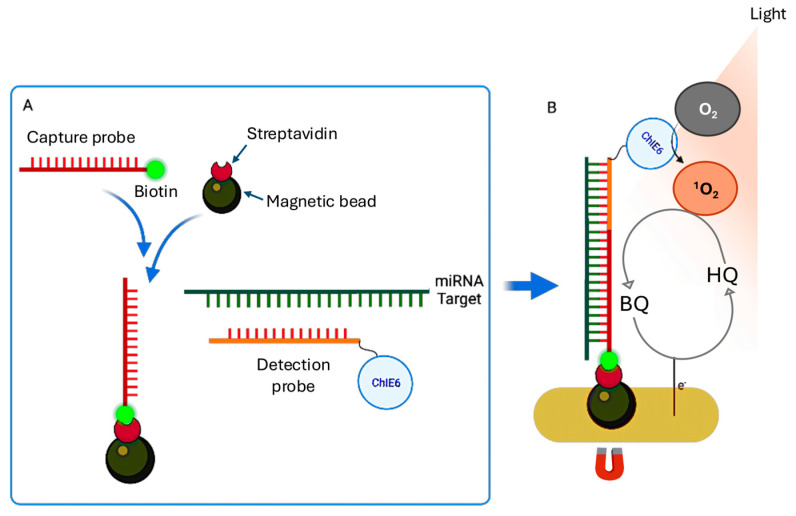
(**A**) Schematic of the sandwich hybridization assay for miRNA detection. The target miRNA is captured between a biotinylated capture probe immobilized on streptavidin-coated magnetic beads and a detection probe labeled with the photosensitizer Chlorin e6 (ChlE6). (**B**) Principle of the (^1^O_2_)-based photoelectrochemical (PEC) detection. Upon light exposure, the photosensitizer generates singlet oxygen (^1^O_2_), which oxidizes hydroquinone (HQ) to benzoquinone (BQ). Figure adapted with permission from [72].

**Figure 7 cancers-17-02715-f007:**
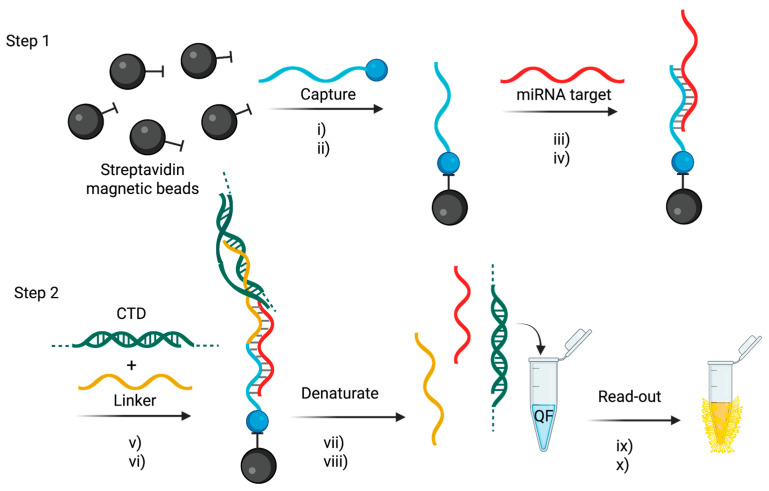
Procedure for the bead-based miRNA detection assay. **Step 1**: (i) Streptavidin-coated magnetic beads were incubated with biotinylated, mutation-specific capture probes (C1–C14) derived from Table 1 [69]; (ii) beads were washed; (iii) incubated with samples containing miR-128-2-3p and its variants; and (iv) washed again. **Step 2**: (v–vi) beads were incubated with CTD, linker, and buffer; (vii–viii) washed and denatured by heating at 92 °C for 10 min; (ix–x) the supernatant was transferred to QuantiFluor (QF) solution, incubated at 92 °C for 5 min, and then cooled to 23 °C over 60 min. Finally, the concentration of miR-128-2-3p and its variants was quantified by fluorescence detection. Figure adapted with permission from [73].

**Figure 8 cancers-17-02715-f008:**
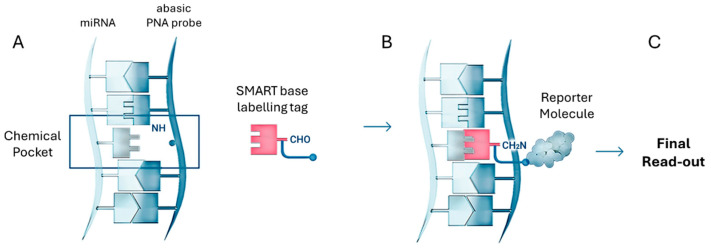
Schematic representation of the DCL process. (**A**) The abasic PNA probe hybridizes with the target miRNA to form a specific chemical pocket. Through a selective covalent reaction, a SMART base carrying a detectable tag is incorporated into this pocket, forming a stable chemical lock. (**B**) This tag is subsequently recognized by a reporter molecule, enabling signal generation and (**C**) Read-out. In most implementations, abasic PNA probes are immobilized on a solid surface, depending on the platform used in conjunction with DCL technology. Common immobilization formats include magnetic beads, sensor chips, or membranes integrated into detection devices. This solid-phase setup enhances hybridization stability, facilitates washing steps and supports high-throughput or multiplexed detection.

**Figure 9 cancers-17-02715-f009:**
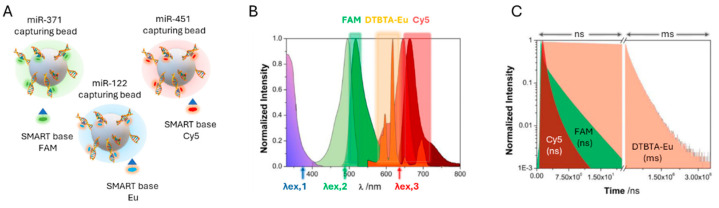
(**A**) Scheme of the multiplexed analysis of miR-371, miR-451 and miR-122. It uses abasic PNA probes in magnetic beads and three different SMART bases tagged with FMA, Eu and Cy5. The different spectra and analysis time windows are shown in panels (**B**,**C**).

**Table 1 cancers-17-02715-t001:** Comparison of novel advanced amplification-free methods for miRNA detection in liquid biopsies.

Technology	Type of Detection	Target miRNA	Limit of Detection (LoD)	Type of Clinical Sample	Types of Cancer	Multiplexing Capability	POCT Suitability	Reference
Solid-state nanoplasmonic sensor	UV-Vis	miR-10b-5p miR-let7a-5p	637.7 aM	Plasma	Pancreatic ductal adenocarcinoma	Yes (can be converted to multiplexed by adding receptors)	Partial (not fully POCT-ready)	[70]
Electro-optical nanopore sensing	Fluorescence	miR-141-3p miR-375-3p	5–8 fM	Serum	Prostate cancer	Yes (up to ~10 targets with potential for more)	Yes	[71]
Singlet oxygen-based photoelectrochemical	Photoelectrochemical	miR-145-5p miR-141-3p	3.5–8.3 pM	Plasma	Prostate cancer	Partial (multiplexing possible with multi-array platforms)	Partial	[72]
Tandem bead-based hybridization assay	Fluorescence	miR-128-2-3p	2.2 pM	Plasma	Colorectal cancer	No (single target focused)	Partial	[73]
Dynamic chemical labeling (DCL)	Fluorescence	miR-21-5p miR-122-5p miR-371a-3p miR-451a-5p	pM range	Serum	Various cancers	Yes	Yes	[74]

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
