# Peer review of "Amplification-Free Testing of microRNA Biomarkers in Cancer"

_cancers, 2025, doi:10.3390/cancers17162715_

Round 1
Reviewer 1 Report
Comments and Suggestions for Authors
Comments:
The review titled “Amplification-Free Testing of microRNA Biomarkers in Cancer” provides a clear and timely overview of emerging amplification-free platforms for miRNA detection. The manuscript is generally well written, but several revisions are suggested to improve its scientific depth and practical relevance. Minor revision is recommended.
- To enhance clarity and conceptual understanding, the authors should add a summary schematic figure that visually integrates the key features of the reviewed amplification-free detection strategies. This will improve the overall readability and facilitate comparison across technologies.
- Please consider citing the following two relevant studies, which offer complementary insights into amplification-free detection strategies: (A) Sensitive detection of microRNAs using polyadenine-mediated fluorescent spherical nucleic acids and a microfluidic electrokinetic signal amplification chip” (https://doi.org/10.1016/j.jpha.2022.05.009). (B) Universal, sensitive, and visual sandwich-type biosensor based on nanogold-catalyzed reduction and its application for detecting C-reactive protein in serum by a portable colorimeter” (https://doi.org/10.1016/j.snb.2025.137408).
- Expand the discussion on clinical translation, including aspects such as sample preparation workflows, degree of automation, and testing cost.
- Revise Table 1 to include more practical evaluation metrics, such as “Multiplexing Capability,” and “POCT Suitability,” to better reflect the translational potential of each method.
- While the DCL platform is important, its coverage is overly detailed compared to the other four methods. This imbalance may affect the neutrality of the review. The authors should streamline the DCL section or provide more analysis of the other platforms.
Author Response
Comment 1: To enhance clarity and conceptual understanding, the authors should add a summary schematic figure that visually integrates the key features of the reviewed amplification-free detection strategies. This will improve the overall readability and facilitate comparison across technologies.
Response 1: We thank the reviewer for this valuable suggestion aimed at improving the clarity and conceptual integration of the manuscript. We acknowledge the potential benefit of including an additional figure to summarise the amplification-free detection strategies visually. However, to maintain visual balance and avoid overcrowding (as the manuscript already includes nine figures), we opted to use Table 1 as the main tool to enhance clarity and support comparison among the five major technologies selected.
We believe that Table 1, now revised and expanded, effectively fulfils this role. In particular, we have improved the summary schematic figure by adding two new columns—“Multiplexing Capability” and “POCT Suitability”—as also suggested by the reviewer in Comment 4. This updated version now provides a more comprehensive visual integration of the key features of each technology while keeping the manuscript concise and reader-friendly.
We hope this solution adequately addresses the reviewer's comment and strengthens the overall impact of the review.
Comment 2: Please consider citing the following two relevant studies, which offer complementary insights into amplification-free detection strategies: (A) Sensitive detection of microRNAs using polyadenine-mediated fluorescent spherical nucleic acids and a microfluidic electrokinetic signal amplification chip” (https://doi.org/10.1016/j.jpha.2022.05.009). (B) Universal, sensitive, and visual sandwich-type biosensor based on nanogold-catalyzed reduction and its application for detecting C-reactive protein in serum by a portable colorimeter” (https://doi.org/10.1016/j.snb.2025.137408).
Response 2: We thank the reviewer for the valuable suggestion and for highlighting two additional relevant works in the field. The section has been revised accordingly to incorporate these studies, which are now properly cited in the manuscript as references 66 and 67. The updated text is highlighted in yellow between lines 342 and 360.
Comment 3: Expand the discussion on clinical translation, including aspects such as sample preparation workflows, degree of automation, and testing cost.
Response 3: We thank the reviewer for this insightful suggestion. In response, the section titled "Conclusion and Future Direction" has been substantially revised to specifically address the clinical translational aspects of the technologies discussed in the review. Particular emphasis has been placed on sample preparation workflows, degree of automation, and testing costs, as recommended.
The revised section provides a more detailed and practical perspective on how these amplification-free technologies could be implemented in clinical settings. The most significant additions have been highlighted in yellow within the manuscript to facilitate review.
We are confident that this expanded discussion enhances the clarity, depth, and clinical relevance of the review.
Comment 4: Revise Table 1 to include more practical evaluation metrics, such as “Multiplexing Capability,” and “POCT Suitability,” to better reflect the translational potential of each method.
Response 4: We thank the reviewer for this constructive suggestion. Table 1 has been revised accordingly to enhance its practical relevance. The updated version now includes two additional columns: “Multiplexing Capability” and “POCT Suitability”, as recommended. These additions provide a clearer overview of each method’s translational potential. The newly added content has been highlighted in yellow to facilitate the review process.
Comment 5: While the DCL platform is important, its coverage is overly detailed compared to the other four methods. This imbalance may affect the neutrality of the review. The authors should streamline the DCL section or provide more analysis of the other platforms.
Response 5: We thank the reviewer for the constructive comment regarding the balance of coverage among the different technologies described in the review. We agree that, in the original version, the section titled "Conclusion and Future Direction" placed disproportionate emphasis on the DCL platform compared to the other technologies.
To address this, we have carefully revised and streamlined the discussion on DCL within the conclusion to ensure a more balanced and neutral presentation. While it is true that DCL represents a well-established platform in the field — with nearly 20 peer-reviewed publications supporting its development and application over the years — we recognise the importance of maintaining an objective tone across all technologies reviewed.
The revised version now gives equal consideration to the broader set of platforms covered, and the updated section has been improved to better reflect their collective translational potential. Changes made in response to this comment have been highlighted in yellow in the manuscript for ease of review.
We hope that this adjustment satisfactorily addresses the reviewer’s concern.
Reviewer 2 Report
Comments and Suggestions for Authors
In this review article, Soleimanpour et al. focus on recent advancements in technologies for detecting circulating miRNAs in clinical samples. First, the authors thoroughly describe several traditional PCR-based methods and their limitations in application for clinical diagnostics. Then, they comprehensively discuss five advanced technologies that can be used for the direct detection of miRNAs circulating in body fluids. The authors explain the advantages and challenges of these novel approaches and the possibility of their applications in the clinic for cancer diagnostics. They highlight Dynamic Chemical Labelling as the most promising method for analyzing circulating miRNAs in liquid biopsy tests.
Given the widely reported abnormal expression of circulating miRNAs in cancer and their potential as biomarkers and/or therapeutic targets, this summary is very useful in deciding the most optimal method for detecting circulating miRNAs in clinical settings.
A few suggestions for better readability:
The section “1. Introduction” is too long and contains an exhaustive description of traditional methods in subsection “1.3. Conventional analytical methods”. Therefore, I would advise presenting subsection 1.3. as a separate section “2. Conventional analytical methods”.
Since the section “2. Aim and methodology” contains an introductory text and Table 1 discussed further in the section “3. Emerging amplification-free methods for circulating miRNA detection in cancer”, I would suggest merging these sections by incorporating the text of section “2. Aim and methodology” at the beginning of section “3. Emerging amplification-free methods for circulating miRNA detection in cancer”.
Minor points:
- The definition of the abbreviation "PNA probes" is not provided.
- Table 1, column “Type of Clinical Sample”: exchange “Sereum” for “Serum”.
Author Response
Comment 1: The section “1. Introduction” is too long and contains an exhaustive description of traditional methods in subsection “1.3. Conventional analytical methods”. Therefore, I would advise presenting subsection 1.3. as a separate section “2. Conventional analytical methods”.
Response 1: We thank the reviewer for this constructive suggestion. Subsection “1.3. Conventional analytical methods” has been moved and presented as a new standalone section, now titled “2. Conventional analytical methods.” The titles and subtitles throughout the manuscript have been modified accordingly to reflect this change. In the revised manuscript, the changes have been highlighted in yellow, and the original subsection title in Section 1 has been marked with a strikethrough to indicate its deletion.
Comment 2: Since the section “2. Aim and methodology” contains an introductory text and Table 1 discussed further in the section “3. Emerging amplification-free methods for circulating miRNA detection in cancer”, I would suggest merging these sections by incorporating the text of section “2. Aim and methodology” at the beginning of section “3. Emerging amplification-free methods for circulating miRNA detection in cancer”.
Response 2: We thank the reviewer for this constructive suggestion. However, we believe it is important to maintain the current structure, as the “2. Aim and methodology” section explains the approach used to select the nine and five final publications, thereby defining the main core of this review. Keeping this section separate ensures clarity in outlining the review’s scope and methodological rigour before presenting the results in the subsequent section. In the revised manuscript, following the changes implemented in response to Comment 1, “2. Aim and methodology” is now presented as Section 3, and the former Section 3 (“Emerging amplification-free methods for circulating miRNA detection in cancer”) is now Section 4. We appreciate the reviewer’s understanding.
3 Minor points:
3.1. The definition of the abbreviation "PNA probes" is not provided.
Response 3.1: We thank the reviewer for noting this omission. In the revised manuscript, the definition of the abbreviation “PNA probes” (Peptide Nucleic Acid probes) has been added at its first occurrence, between lines 355 and 356, and highlighted in yellow.
2. Table 1, column “Type of Clinical Sample”: exchange “Sereum” for “Serum”.
Response 3.2: Change has been made. In Table 1, column “Type of Clinical Sample,” the term “Sereum” has been corrected to “Serum.”
Reviewer 3 Report
Comments and Suggestions for Authors
The manuscript presents, mainly, an overview of five еmerging methods for detecting circulating microRNAs in cancer that do not require amplification. The review will be useful to a wide range of researchers working in this promising and in-demand field.
The manuscript may be published as is.
However, since the first and third authors are affiliated with a company promoting a variant of the ChemiRNA™ Tech (https://destinagenomics.com) that is most favorably presented in the review as Dynamic Chemical Labeling (DCL) technology, it is recommended to clearly indicate this in the Conflict of Interest section.
In addition, typos, for example, such as SNART in line 666 should be corrected.
Author Response
Comment 1: However, since the first and third authors are affiliated with a company promoting a variant of the ChemiRNA™ Tech (https://destinagenomics.com) that is most favorably presented in the review as Dynamic Chemical Labeling (DCL) technology, it is recommended to clearly indicate this in the Conflict of Interest section.
Response 1: We thank the reviewer for this observation. In the revised manuscript, a full dedicated paragraph has been added to the Conflict of Interest section, clearly indicating that the first and third authors are affiliated with a company promoting a variant of the ChemiRNA™ Tech, most favorably presented in the review as Dynamic Chemical Labeling (DCL) technology. This paragraph is highlighted in yellow and appears between pages 736–740.
Comment 2: In addition, typos, for example, such as SNART in line 666 should be corrected.
Response 2: We thank the reviewer for noting this. The typo “SNART” has been corrected in the revised manuscript. This correction is highlighted in yellow at line 661.